# Healthcare Waste and Sustainability: Implications for a Circular Economy

Abrar Mahjoob *, Yousef Alfadhli * and Vincent Omachonu

Department of Industrial and Systems Engineering, University of Miami, Coral Gables, FL 33124, USA; vomachonu@miami.edu
* Correspondence: afmahjoob@kau.edu.sa (A.M.); yxa403@miami.edu (Y.A.)

**Abstract:** The field of healthcare waste systems is an emerging research field with new methodologies being developed to reinforce sustainability. Medical treatments are becoming more sophisticated and in demand due to increasing incidences of chronic disease. Healthcare is also becoming widely available worldwide. Healthcare waste disposal includes multiple disposal methods including incineration, landfilling, and chemical treatments. These rudimentary methods and their increased prevalence present their own problems that negatively impact both the environment and public health. As a result, there is a global call for healthcare waste systems to transition from a linear to a circular economy (CE). The CE philosophy is employed in other waste management industries. There are limited studies, however, that assess the challenges to CE implementation, particularly in the United States. This research presents the challenges to the implementation of a CE in healthcare waste management systems. The challenges were determined by an extensive literature review. Three experts from the industry verified and provided additional context to the challenges through meetings. The challenges were grouped into seven categories: supply chain management, regulations and policies, industry, organizations internal, technology and operational, economic, and funding, and social. A decision-making trial and evaluation (DEMATEL) was used to rank the challenges and illustrate interrelationships between the challenges. The most significant challenge was found to be a lack of governmental legislation on CE healthcare policies, followed by a lack of a realistic CE business model and lack of consumer interest in the environment. The research will provide context to further developments into adopting CE practices. The limitations and future scope of the research are presented.

**Keywords:** healthcare waste management; circular economy; DEMATEL method





## 1. Introduction

As it stands, healthcare is one of the fastest growing global industries in recent years [1], with worldwide health spending rising higher than ever and continuing to rise every year. In 2017, the global health spending was estimated to be around 7.8 trillion USD worldwide and was 10% of the worlds gross domestic product (GDP) [2]. The increase is attributed to the population increasing and thus, there is a greater need for healthcare interventions [3]. The global population is increasing and is set do so, with an estimate of 8.6 billion by 2030 and 9.8 billion by 2050 [4]. With populations aging, and life expectancy increasing, chronic disease spending continues to grow, as well as the cost of having multiple morbid conditions (MMCs) [3]. MMCs lead to elevated use of primary care and specialist care services, in addition to increased medication use and hospital admissions [5].

In the United States (US), pollution from the healthcare industry results in up to 614,000 disability-adjusted life years (DALYs) lost annually [6]. The health sector is responsible for 4.6% of global greenhouse gas emissions, more than a quarter of which is rooted from within the US healthcare system. The bulk of healthcare global greenhouse gas emissions are caused by the supply chain, making it an area for highest impact on health

care decarbonization [7]. During the past thirty years, the healthcare industry—particularly industries in high-income and developed nations—has become heavily reliant on rudimentary waste disposal methods and single-use disposable medical devices [8].

Industrial economies, although evolving significantly throughout history, have never deviated from a resource-based linear model. Single-use disposables are symbolic of a linear economy which follows a sequence of 'take-make-dispose'. Products are manufactured, used once and then disposed. Although linear economies have flourished, they come with a cost to society and public health. The linear model is intrinsically unsustainable as mass production and consumption contribute to the global ecological destruction by generating excessive solid waste, greenhouse gases, and other detrimental emissions [8]. In addition, their inordinate use of raw materials and production of goods have led to harmful effects on the environmental ecosystem [9]. These effects endanger public health by air pollution, water and soil contamination, ozone depletion, biodiversity loss and catastrophic climate change [8]. Industries extract materials and use energy and labor to produce goods that are sold to customers. These goods are then either incinerated, sent to landfills, or are chemically treated where they no longer serve a purpose [10].

The disadvantages associated with linear economies provide incentive to shift towards a circular economy (CE), as it is a more sustainable framework. CE is defined as 'an economic system that uses the reuse of products and materials as a starting point, as well as the conservation of natural resources, where economic, social, and environmental values are relevant in all aspects of the system [11,12]. The philosophy combines ideas of industrial ecology, natural capitalism, biomimicry, and performance economy. In essence, a CE minimizes waste and maximizes resource productivity and builds resilient supply chains and reinforces social values [8].

The growing epidemic of chronic diseases has led to healthcare waste (HCW) disposal practices being utilized extensively, and thus the HCW disposal industry is growing. Environmental impact on public health has been one of the essential areas to save humanity, earth and medical waste that has been generated from all kinds of medical sectors, especially since COVID-19 began threatening public health and quality of life. The volume of waste generated during COVID-19 has provided various opportunities to implement the principles of a CE.

HCW is a matter of great concern for the environment and public health due to its infectious and hazardous nature [13]. The existing studies in the domain of implementation in HCW are limited to suitable treatment methods for the safe disposal of HCW [13]. Thus, developing an extensive framework to evaluate the challenges associated with the implementation of CE is a gap in research. The evaluation would allow for advancements in the recycling and recovery of HCW. It would also aid in the drafting of national plans for minimizing the waste generated and implementing the 4Rs (Reduce, Reuse, Recycle, Recover) for HCW. In addition, it would aid in the realistic design of closed loop supply chains [14].

A circular economy is restorative and regenerative in contrast with a linear economy, which regards energy resources as disposable [8]. The main principle of CE is to protect the value and quality of products/materials by extending the end-of-life cycle of the product [15]. The presence of challenges and lack of suitable infrastructure to achieve a sustainable HCW system have been cited as a challenge [13]. Furthermore, the current literature in the HCW sector shows a gap in the connection between CE and multi-criteria decision-making models, particularly in developed economies. The following are the objectives of the research paper:

1. To identify the challenges to implementing a CE in HCW.
2. Develop a hierarchal framework to evaluate these challenges.
3. Create a foundation for future research in HCW management in the context of a circular economy.

To address the objectives, the current literature on HCW management and the concept of a CE were analyzed. The literature review paved the way to develop an extensive framework of challenges and their respective sub-challenges to a CE in the healthcare sector.

Next, the framework was evaluated with the aid of expert opinion in the Miami HCW sector to refine the sub-challenges. Following this, the decision-making trial and evaluation laboratory method (DEMATEL), a multi-criteria decision-making model (MCDM), was applied to identify the weights of challenges. The weighting allowed for further insight into which challenges are significant. Additionally, cause and effect relationships between challenges are illustrated. In this context, the main contribution of this research is the evaluation of the current HCW and CE adoption literature and the proposal of a MCDM-enabled framework that ranks challenges to CE, as well as identifying cause and effect relationships. The aim is to establish a foundation for future research in HCW management in the context of a CE by realistically identifying the barriers to sustainability.

The paper is structured with an initial literature review to evaluate the challenges to CE implementation, and an assessment of recent literature in the field of CE regarding HCW management. Following is the theoretical background on the CE challenges. Next, the DEMATEL methodology is applied, followed by a discussion of the findings. To conclude, the theoretical implications, limitations and concluding remarks are presented.

## 2. Literature Review

### 2.1. Circular Economy in Healthcare Waste Management

Environmental sustainability and the transition from a linear economy to a CE heavily relies on effective waste management. CE, with a focus on the waste hierarchy from waste prevention at the top to disposal at the bottom, aims to close the supply chain loop as much as possible, to allow for a sustainable and zero-waste environment [8]. The management of waste plays a significant role in environmental health and the transition to a CE. Thus, the design and management of efficient waste management systems are the foundation to setting up a CE economy [16]. A CE minimizes resource input waste and energy leakage by slowing and closing material and energy loops [17,18]. Slowing loops refers to maximizing a product's lifecycle. Closing loops refers to creating value from waste by finding new applications for it [8].

Figure 1 illustrates where the circular action can take a place in the supply chain. Reuse action can be implemented between the disposal and healthcare provider stage. For example, reusable surgical gowns are generally FDA-approved for 75 reuse cycles before they are no longer suitable for protection. Life-cycle assessments demonstrate that reusable gowns generate less solid waste and half the amount of greenhouse gas emissions when compared to disposable gowns [19].

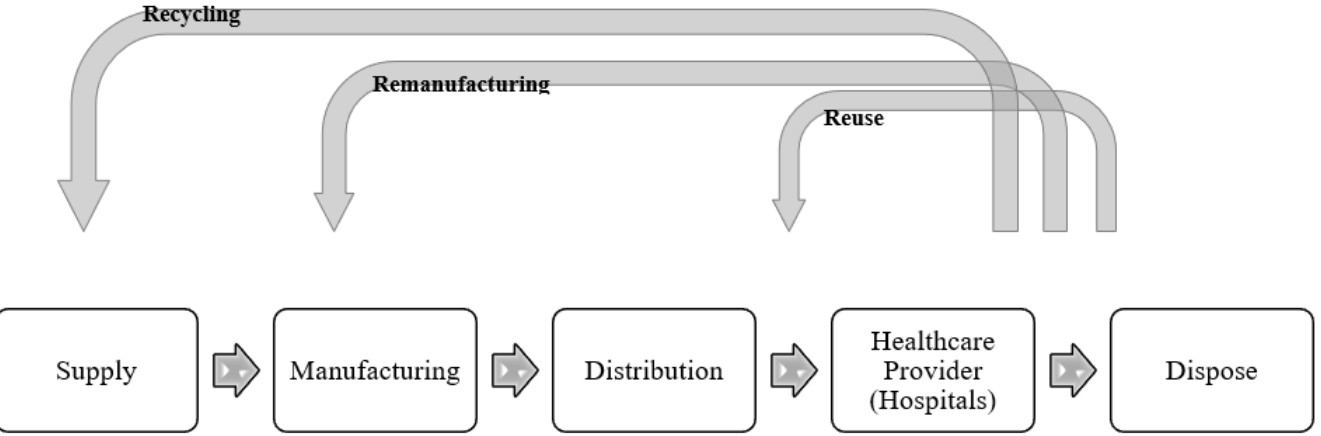

**Figure 1.** Circular Economy Stages.

Reusing surgical gowns in this case would significantly benefit the environment and reduce reliance on disposables. When the surgical gowns reach the end of their life cycle, they can be used in other industrial applications, such as insulation [8]. Remanufacturing action can happen between the disposal and manufacturing process. Many single-use

disposables can be safely reprocessed; however, hospitals have avoided doing so due to liability and cost concerns. Recently, third-party entities have emerged to allow hospitals to outsource disposables for remanufacturing. The remanufacturing process extends the life cycle of products, which results in cost savings for the hospitals. More importantly, remanufacturing reduces pollution from excessive production and raw material consumption. According to a report by the Association of Medical Device Reprocessors (AMDR), remanufacturing saved U.S. hospitals $372 million in 2020. Remanufactured devices cost 25% to 40% less, and in consequence, the reduction of medical waste further saves costs. The report also states that if remanufacturing practices of the top 10% performing hospitals were emulated nationally, U.S. hospitals could save an additional $2.28 billion in 2020 [20]. The recycling action is where the raw material goes back to the supplier for recycling purposes, which is typically the lowest yield solution to reduce waste.

To further illustrate the closed supply chain loop, Figure 2 illustrates the CE cycle. Waste management includes all activities and actions required to manage waste from its inception to its final disposal though the collection, transport, and treatment phases [21]. Appropriately managing, mitigating and valorization of waste are essential to transform our society to a zero-waste and sustainable environment. Waste and pollution prevention are the key reasons for developing a CE. Policy makers embrace the zero-waste concept, however, there's been a notable lack of advanced research in the domains of zero waste design and evaluations [21].

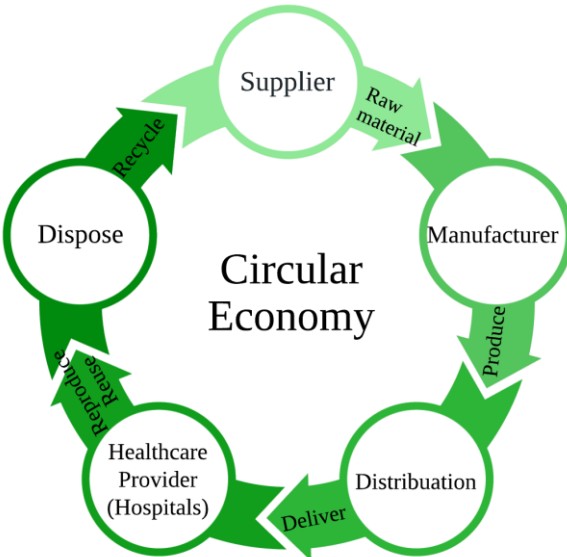

**Figure 2.** Circular Economy Cycle.

The goal of a healthcare service provider is to attain the best services for every patient's need, which can only be achieved by implementing the highest standard practices in every domain within the industry. High-value care is defined by the elimination of unnecessary waste and inefficiency, while maximizing patient outcomes and experience [22]. Moreover, HCW management and disposable practices are essential to improve to sustain the environmental impact and reach the concept of the circular economy. The healthcare sector has been facing several challenges, especially during COVID-19. The challenges demonstrated that single-use disposables are convenient under normal day-to-day conditions, however, heavily relying on them leads to failure [8]. These challenges could be concluded in the concept of the Triple Bottom Line (TBL); economic, environmental, and social. As attractive as the incentives for shifting to a CE, doing so will require a systemic transformation by all stakeholders.

## 2.2. Challenges of Circular Economy

A systematic literature review was conducted to gain deeper insight into what challenges are hindering the shift to a CE. Seven major challenges were identified and split into their respective sub-challenges: supply chain management (C1), regulations and policies (C2), industry (C3), organizations internal (C4), technology and operational (C5), economic and funding (C6) and social (C7). Table 1 outlines the main challenges and their respective sub-challenges. Table 2 provides deeper insight to a description of the respective sub-challenges.

**Table 1.** Identified main challenges and sub-challenges in adoption of CE practices.

| Main Challenges | Code | Sub-Challenges | Code |
|---|---|---|---|
| Supply Chain Management Challenges | C1 | Uncertainty in return flows | S1 |
| | | Lack of consideration for a CE supply chain | S2 |
| | | Inefficient resource utilization | S3 |
| | | Lack of flexibility in implementation of CE phases | S4 |
| | | Challenges in taking back products (4Rs) | S5 |
| Regulation and Policies Challenges | C2 | Lack of governmental legislation on CE healthcare policies | R1 |
| | | Lack of sustainability foundation practices, incentives and policies | R2 |
| | | Lack of R&D to reinforce sustainability practices and create/update new legislation | R3 |
| | | Lack of standards for existing CE | R4 |
| Industry Challenges | C3 | Resistance to improve existing practices to be sustainable | I1 |
| | | Minimal knowledge on training and expertise for sustainability practices | I2 |
| | | Lack of realistic CE business model | I3 |
| | | Lack of consumer interest in the environment | I4 |
| Organization's Internal challenges | C4 | Carelessness in usage | O1 |
| | | Poor contribution by stakeholders (information exchange) | O2 |
| | | Strict requirements for advanced education and training | O3 |
| | | Conflict of interests among departments | O4 |
| Technology and Operational Challenges | C5 | Lack of environmentally friendly disposal practices | T1 |
| | | Challenges in reprocessing | T2 |
| | | Insufficient product traceability | T3 |
| | | The associated complexity with circular economy practices | T4 |
| | | Lack of key performance indexes to adequately measure "green" performance of hospitals | T5 |
| | | Minimal advocation to use medical informatics to reinforce sustainability | T6 |
| Economic/Funding Challenges | C6 | Low eco-efficiency of technological processes | E1 |
| | | Conflict of interests and poor communication among departments | E2 |
| | | High costs associated with circular economy disposal practices | E3 |
| | | Unwillingness to invest more | E4 |
| | | Inadequate allocation of funds | E5 |
| Social Challenges | C7 | Cultural barriers | SO1 |
| | | Minimal public incentives and interest regarding sustainability practices | SO2 |
| | | Lack of awareness about CE practices and resources | SO3 |
| | | The widespread use disposal of medical products/devices/supplies | SO4 |
| | | Lack of environmental impact awareness on public health | SO5 |

**Table 2.** Identified sub-challenges descriptions.

| Code | Description | References |
|---|---|---|
| S1 | Uncertainty in the quality, quantity, timing, and location of end-of-use returned products makes capacity planning challenging. | [23] |
| S2 | Minimal knowledge on adapting current supply chains: closing the supply loop to make it more efficient. | [24] |
| S3 | Inaccuracies in forecasting. Ordering items solely based on what is currently needed instead of for the future. | [25] |
| S4 | Inflexible infrastructure and processes to improve current methods to be more sustainable. (Segregation, transportation, disposal). | [15] |
| S5 | Reduce | Re-use | Recycle | Recover is challenging. | [26] |
| R1 | Governmental legislation is needed to enforce organizations transitions to CE to achieve advantages in terms of resource use efficiency. | [27] |
| R2 | Government should emphasize and incentivize hospitals to adopt CE practices to reduce public health impact. And, to save the hospitals money. | [8] |
| R3 | Lack of studies in the CE field emphasizing the importance of transitioning to a CE. | [28] |
| R4 | As CE is an emerging field, metrics for measuring a hospitals performance are generally archaic and not in line with sustainable technologies. | [29] |
| I1 | Changes to be more sustainable have implications (cost, time, effort, etc.) and thus there is a resistance to shift practices. | [30] |
| I2 | Not enough training programs in healthcare sustainability emphasizing CE waste management. | [31] |
| I3 | A business model would ease the process on hospitals. Unavailability of proper CE business models creates problems. | [24] |
| I4 | If consumers aren't interested enough to protect the environment, implementation of CE is difficult. | [27] |
| O1 | Consuming items without considering consequences. Over-use without taking responsibility. | [25,32] |
| O2 | Information exchange between hospitals stakeholders and government representatives in order to adopt sustainable healthcare systems. | [8] |
| O3 | No incentives to educate and train extensively (financially, socially, etc.) | [33] |
| O4 | The easiest methods are generally the most attractive and least sustainable. | [34] |
| T1 | Poor responsiveness from hospitals to adapt more "green" waste management methods. | [3] |
| T2 | Hospitals avoid reprocessing due to the liability issues, maintenance, cost, approval for reprocessed products. | [35] |
| T3 | Information systems are generally insufficient, and the inability to trace products makes it difficult to collect and refurbish products. | [30] |
| T4 | Minimal technological product infrastructure. Current technological infrastructure is difficult to adapt to shifting towards CE practices. | [15] |
| T6 | A CE focus requires a shift from a solely volume drive economic approach to consider triple bottom line (economic, environmental, and social factors) | [29] |
| T7 | In general, current practices rely on vintage techniques to trace medical waste. The use of digital technology could be beneficial from a sustainability perspective. | [36] |
| E1 | Recycling operations may be expensive and inefficient, which result in material loss and cross contamination. | [37] |
| E2 | CE needs efficient coordination and information exchange throughout all departments, which is difficult to implement due to budgeting limitations. | [30] |
| E3 | Implementing CE increases manufacturing cost due to the urgency to reuse/recycle products. | [38] |
| E5 | Always the consumer-aim is to reduce cost when pricing is a key factor for profitability. There is a cost associated with CE due to reprocessing/reusing for some products | [31] |
| E6 | Poor financial planning leads to lack of investment in CE practices. Which is an area to focus on to reduce environmental impact. | [31] |
| SO1 | Conflict in medical waste management practices due to backgrounds and cultural differences. Some cultures are accustomed to disposing of things immediately, while others may not be. | [39] |
| SO2 | Not enough social incentive to be more "green" due to many factors: education, funding, and public awareness. | [40] |
| SO3 | Lack of education and resources available regarding the new trends in waste management to reinforce sustainability. | [41] |
| SO4 | Due to the pandemic, much research has been done in waste management practices to negate the problems associated with single use products. | [42] |
| SO5 | Lack of consumer interest in reducing environmental impact of improper waste management techniques. | [43] |

### 2.2.1. Supply Chain Management Challenges (C1)

A CE supply chain relies on a well-built coordination and exchange of information among all levels of the supply chain [44]. At the foundation of it, there is minimal knowledge on adapting current supply chains to be more circular [24]. There are also clear problems in forecasting, where items are being ordered as required rather than by demand [25]. The discussion of CE is generally limited to corporate social responsibility and environmental departments while having much less appeal to influential departments such as operations and finance. In addition, the lack of flexibility in implementation is partly due to the lack of infrastructure to adopt CE practices. CE would incur great costs initially with increased transportation and operation costs for recycling or remanufacturing [15]. The lack of infrastructure also makes it challenging to take products to re-use, remanufacture or recycle [26].

### 2.2.2. Regulations and Policies Challenges (C2)

Systemic transformation can rely heavily on the incentives present to do so. Implementation of a CE is considered at two levels. Macro-level strategies involve nationwide implementation such as the European Union's Circular Economy Action Plan and China's Circular Economy Promotion Law, while Micro-level implementation focuses on a group of sectors [8]. Regulations and incentive policies are powerful tools to encourage business models to adapt CE practices. Government plays a key role in creating these incentives to enforce organizations' transitions to CE to achieve advantages in terms of resource efficiency and cost savings [27]. In addition, CE needs a shift from focusing on the total amount of waste generated and its economic repercussions, to one that considers environmental and social factors as well [29].

### 2.2.3. Industry Challenges (C3)

At the core of the HCW industry, there is clear resistance to transformation due to the cost, time, and effort implications [30]. The lack of a realistic CE business model creates challenges, as without one, the industry has no economic incentive to transform [24]. In addition, particularly in developing countries, there is a notable lack of training and education on healthcare sustainability that emphasize CE practices [31]. Moreover, there simply is not any real incentive for consumers to care for the environment [27].

### 2.2.4. Organizations Internal Challenges (C4)

The shift to a CE relies on collaboration between all stakeholders in the HCW industry. Information exchange between governments, hospitals, and third-party entities is crucial to adopt sustainable HCW practices [8]. A significant challenge is the barrier of entry to work in the HCW sector. Extensive training is required, with thorough background checks and social responsibility with no real financial incentive to do so [33]. In addition, internally there is a prevalence among consumers to over consume items without considering the consequences. Patient care and outcome is of the highest importance, and achieving the highest standard requires all stakeholders to consider environmental and social implications of HCW management [25,32].

### 2.2.5. Technology and Operational Challenges (C5)

Effective HCW management is built on the technology available to carry out the processes. The technological infrastructure currently relies heavily on rudimentary methods of incineration or landfilling to dispose of waste, with a notable lack of environmentally friendly disposal methods [3]. Transforming the technology to follow a CE approach is difficult [15]. In addition, the prospect of remanufacturing is one that hospitals typically avoid due to liabilities, cost, maintenance and approval issues [35]. Information systems to trace medical waste are generally inefficient. The inability to trace products makes it difficult to remanufacture them to extend their life cycle [30].

### 2.2.6. Economic/Funding Challenges (C6)

The transition from a linear to CE demands a significant initial investment. The costs associated with reprocessing, recycling, or re-using waste makes the option unattractive. In addition, recycling operations can be expensive and inefficient, leading to material loss and cross contamination [37]. Stakeholders gravitate towards minimizing costs, and the easiest options to dispose of HCW are typically the least expensive and least environmentally friendly. As a result, there is an unwillingness by stakeholders to invest more [31]. In addition, conflicts of interests among departments leads to inefficient information exchange and coordination with budgeting [30]. Lack of investment into CE infrastructure can be linked to the conflict of interest with budgeting, and the inadequate allocation of funds to reduce environmental impact [31].

### 2.2.7. Social Challenges (C7)

Cultural barriers present themselves at the forefront of social challenges when it comes to a CE. In HCW, some cultures may be predisposed to disposing of waste in a certain way that can be over or under what is required by protocol [39]. Training and education are core tools to reduce waste and improve eco-efficiency of the HCW processes. However, there has been a notable lack in education available regarding the new trends in waste management to reinforce sustainability [41]. There is minimal social incentive to adopt green practices due to lack of awareness on environmental impacts associated with waste mismanagement [34,40]. Furthermore, the widespread use of disposable medical devices, particularly during COVID-19, and the mismanagement of waste then is a significant challenge. A significant amount of research has been conducted on waste management practices to negate the negative effects during the pandemic [42]. The pollution because of the pandemic and its implications on the environment and public health will become prevalent in the coming years [3]

### 2.3. Recent Literature Related to Healthcare Waste Sustainability

Since the COVID-19 pandemic, a significant amount of research has been conducted in the field of HCW management. Table 3 outlines the studies in the field of HCW, specifically in the context of a CE. A study by Patil et al. [45] investigated medical and pharmaceutical waste management using the fuzzy-best worse methodology. The study found that the largest barrier to sustainability was material issues, and the complexity associated with remanufacturing. Kazancoglu et al. [46] identified barriers to CE in the healthcare sector using fuzzy-best worse method. The study found that the high-cost requirements for CE technologies were a significant barrier. However, the study states its limitation in its context, as it is conducted in a country with a developing economy. Karuppiah et al. [47] identified challenges hampering sustainable humanitarian supply chains during the COVID-19 pandemic. Facility location problems and the short lead time for supplies were found to be significant challenges. Kandasmy et al. [41] investigated CE adoption challenges using fuzzy theory and found that the most significant group of challenges relates to supply chain and the available technology. However, the study states that it is limited by being conducted in a country with a developing economy. Ranjbari et al. and Ghisellini et al. [14,48] conducted thorough literature reviews on the philosophy of a CE in all waste management industries. In the context of inventory management systems, Sohrabi et al. [49] proposed a dynamic demand centered framework for managing a blood bank's inventory. The authors used object oriented and integrated computer aided manufacturing approach to develop their process. The model developed can provide the data needed to create a blood logistic plan and its respective operational infrastructure. However, the study states its main limitation lacks an optimized mathematical model alongside the framework itself.

**Table 3.** Recent research studies on healthcare waste.

| | Objectives | Methods | Outcomes | Limitations | Sources | Journal |
|---|---|---|---|---|---|---|
| 1 | Investigate medical and pharmaceutical waste management | Fuzzy best-worse | Material issues are the largest barrier to sustainability. | Lack of diversity in respondents | [45] | Sustainable Production and Consumption |
| 2 | Identify barriers related to Circular Economy in Healthcare Sector. | Fuzzy best-worse and Fuzzy Vikor | High-cost requirement for circular technologies and implementations was found to be most important barrier. | Study conducted in a country with a developing economy. | [46] | International Journal of Environmental Research and Public Health |
| 3 | Identify and evaluate challenges hampering sustainable humanitarian supply chain management during COVID-19. | AHP and TODIM | Facility location problems, short lead times for supplies, spread of rumors, emergence of new clusters, doubt concerning vaccine are critical challenges. | Study conducted in India only and based on local expert feedback. | [47] | Sustainability |
| 4 | Investigate circular economy adoption challenges. | Fuzzy theory | Most significant group of challenges related to supply chain management and technology. | Study conducted in a country with a developing economy. | [41] | Wiley |
| 5 | Provide inclusive map of background of Waste Management and Circular Economy over the last two decades. | Mixed-Method Approach | Identified seven major research themes. | Research themes clustered based on bibliographic coupling of articles. | [14] | Journal of Cleaner Production |
| 6 | Provide review of literature on main CE features. | Literature Review | | Future evaluation and monitoring needed on CE principles applied. | [48] | Journal of Cleaner Production |
| 7 | Propose a dynamic demand-centered framework for managing a blood banks inventory. | Object Oriented and Integrated Computer Aided Manufacturing (ICAM). | Developed a model that captures top features of the inventory system to understand the systems elements and attributes. | Future research requires a mathematical optimization model alongside the framework, | [49] | Healthcare Informatics Research |

## 3. Methodology

The aim of this study is to prioritize the identified main group of challenges and their sub-challenges using the Decision-Making Trial and Evaluation Laboratory Method (DEMATEL). The main challenges and sub-challenges were identified from a previous literature review and modified by experts in the field. Seven main challenges were finalized along with their sub-challenges. Expert opinion was used as an input for conducting DEMATEL. The result will assist in obtaining the impact relationship map and identifying cause-effect relationships between the main group of challenges and sub-challenges. The detailed process of the DEMATEL is presented in the following subsections.

### 3.1. Data Collection

Experts in the field assist in developing the main challenges and sub-challenges, providing their insights and comments reflecting on the reality of the situation. Two valuable meetings were conducted with each expert. The main source of data collection used as an input for DEMATEL is consulting the experts in the field. The data from these meetings was collected in a Microsoft Excel spreadsheet.

### 3.2. Expert Selection

Insights gained from the experts aid in validating and finalizing the main group challenges and their respective sub-challenges after the literature review phase. This step assists the study to reflect the reality of the situation against environmental sustainability. Table 4 illustrates the expert participants' background.

**Table 4.** Information about Expert Participants.

| Expert | Education Level | Position of the Participants | Years of Work Experience | Hospital Specialty |
|---|---|---|---|---|
| 1 | Bachelor's degree | Environmental Director (Environmental services, EV) | More than 20 years | General hospital |
| 2 | Bachelor's degree | Director of Facilities & Operations | More than 20 years | General hospital |
| 3 | Master's degree | Assistant Vice President (AVP), Support Services | More than 10 years | Independent non-profit hospital |

### 3.3. Decision Making Trial and Evaluation Laboratory Method (DEMATEL) Steps

The Geneva Research Centre of the Battelle Memorial Institute introduced the decision-making trial and evaluation laboratory (DEMATEL) technique to the research community to illustrate the complicated relationships with the aid of matrices or digraphs. This approach shows how the criteria are interdependent. The DEMATEL framework is built on the premise that all criteria are interdependent and affect one another [50]. The opinions of experts in the field gathered to rank the criteria that are stated in Tables 2 and 3.

Following that, the scale of measuring the interdependence between each criterion that is used to compile the decision-maker's responses is shown in Table 5. This indicates that all the diagonal elements are zero because the influence of criteria over itself is zero. However, from 1 to 4 indicates there is an influence between the intersected criteria, going from very low to very high influence [50].

**Table 5.** Pair wise Comparison scale of the DEMATEL method.

| Intensity of Value | Interpretation |
|---|---|
| 0 | No influence |
| 1 | Very low influence ($x_i$ has very low influence over $x_i$) |
| 2 | Low influence ($x_i$ has low influence over $x_i$) |
| 3 | High influence ($x_i$ has high influence over $x_i$) |
| 4 | Very high influence ($x_i$ has very high influence over $x_i$) |

The following explained steps of the DEMATEL have been implemented as same as the structure of the [41] study with some modifications.

Step 1: Collect expert opinions and formulate the individual direct-relation matrix (X).

Equation (1) used to collect the individual direct-relation matrix using the experts' judgments according to their level of experience. Each expert was requested to assign a score in terms of intensity of value shown in Table 5.

The individual direct relation matrix denotes as (X) a $n \times n$ positive matrix, for each expert's data input is:

$$X^k = [x_{ij}] \tag{1}$$

where $i = 1, 2, 3, \ldots, n$, $j = 1, 2, 3, \ldots, n$, and $k = 1, 2, 3, \ldots, e$ ($e$: number of experts).

Step 2: Calculate the average direct-relation matrix (A).

In general, it is challenging to consider all perspectives when there are multiple opinions for a decision. Consequently, since there are no decision-makers in a group decision-making situation. Equation (2) shows the decision-making process starts with the average value. The average matrix for the main challenges is presented in Table 6. The matrix consists of seven main challenges as defined in Table 1. An acronym ($C_i$) is given for

each main challenge where *i* is an index for these challenges in the order of Table 1. Where *i* = 1, 2, 3, 4, 5, 6, 7.

**Table 6.** Average direct-relation matrix or average matrix of the main groups of challenges.

|      | C1 | C2 | C3 | C4 | C5 | C6 | C7 |
|------|----|----|----|----|----|----|----|
| C1   | 0  | 2  | 2  | 2  | 2  | 2  | 2  |
| C2   | 2  | 0  | 3  | 2  | 3  | 2  | 2  |
| C3   | 2  | 2  | 0  | 3  | 2  | 2  | 2  |
| C4   | 2  | 2  | 3  | 0  | 2  | 2  | 2  |
| C5   | 2  | 2  | 2  | 2  | 0  | 2  | 2  |
| C6   | 2  | 2  | 2  | 2  | 2  | 0  | 2  |
| C7   | 1  | 2  | 2  | 1  | 2  | 2  | 0  |

Abbreviations: C1, supply chain management challenges; C2, regulation and policies challenges; C3, industry challenges; C4, organization's internal challenges; C5, technology and operational challenges; C6, economic/funding challenges; C7, social challenges.

The average direct-relation matrix or average matrix denotes as (*A*) a *n* × *n* positive matrix, for all experts is:

$$A = \frac{1}{n} \sum_{k=1}^{n} X_{ij}^{k} \tag{2}$$

where *i* = 1, 2, 3, . . . , *n*, *j* = 1, 2, 3, . . . , *n*, and *k* = 1, 2, 3, . . . , *e* (*e*: number of experts).

Step 3: Normalizing the average direct-relation matrix (*N*).

In Table 7, The normalized average direct-relation matrix *N* = [$n_{ij}$] is calculated in this step. Equations (4) and (5) were used to compute *N*. The normalized direct-relation matrix is applied to main challenges and sub-challenges. The calculated normalized average matrix is presented in Table 7.

$$N = \begin{bmatrix} n_{11} & n_{12} & n_{1n} \\ n_{21} & n_{22} & n_{2n} \\ n_{n1} & n_{n2} & n_{nn} \end{bmatrix} \tag{3}$$

$$R = \max \left( \sum_{i=1}^{n} a_{ij}, \sum_{j=1}^{n} a_{ij} \right) \tag{4}$$

$$n_{ij} = \frac{a_{ij}}{R} \tag{5}$$

where *i* = *j* = 1, 2, 3, . . . , *n*.

**Table 7.** The Normalized average direct-relation matrix of the main groups of challenges.

|      | C1    | C2    | C3    | C4    | C5    | C6    | C7    |
|------|-------|-------|-------|-------|-------|-------|-------|
| C1   | 0     | 0.079 | 0.079 | 0.095 | 0.095 | 0.079 | 0.095 |
| C2   | 0.079 | 0     | 0.127 | 0.111 | 0.127 | 0.095 | 0.095 |
| C3   | 0.079 | 0.111 | 0     | 0.143 | 0.111 | 0.095 | 0.111 |
| C4   | 0.079 | 0.095 | 0.127 | 0     | 0.111 | 0.095 | 0.095 |
| C5   | 0.079 | 0.111 | 0.095 | 0.111 | 0     | 0.095 | 0.111 |
| C6   | 0.079 | 0.095 | 0.095 | 0.095 | 0.095 | 0     | 0.095 |
| C7   | 0.063 | 0.079 | 0.079 | 0.063 | 0.079 | 0.079 | 0     |

Abbreviations: C1, supply chain management challenges; C2, regulation and policies challenges; C3, industry challenges; C4, organization's internal challenges; C5, technology and operational challenges; C6, Economic/Funding challenges; C7, Social challenges.

Step 4: Calculate the total-relationship matrix (T).

Table 8 presents the result of the total-relationship matrix (*T*). The total-relationship matrix (*T*) is computed using Equation (6) by using the calculated parts in the presented step 3.

$$T = N(I - N)^{-1} \tag{6}$$

**Table 8.** The total-relationship matrix of the main groups of challenges.

|  | **C1** | **C2** | **C3** | **C4** | **C5** | **C6** | **C7** |
|---|---|---|---|---|---|---|---|
| C1 | 0.088 | 0.182 | 0.187 | 0.203 | 0.203 | 0.176 | 0.200 |
| C2 | 0.180 | 0.131 | 0.250 | 0.241 | 0.253 | 0.211 | 0.224 |
| C3 | 0.182 | 0.233 | 0.140 | 0.269 | 0.243 | 0.213 | 0.239 |
| C4 | 0.175 | 0.212 | 0.244 | 0.134 | 0.233 | 0.205 | 0.217 |
| C5 | 0.174 | 0.224 | 0.217 | 0.232 | 0.133 | 0.204 | 0.229 |
| C6 | 0.167 | 0.202 | 0.208 | 0.210 | 0.210 | 0.109 | 0.207 |
| C7 | 0.136 | 0.167 | 0.171 | 0.160 | 0.174 | 0.162 | 0.098 |

Abbreviations: C1, supply chain management challenges; C2, regulation and policies challenges; C3, industry challenges; C4, organization's internal challenges; C5, technology and operational challenges; C6, Economic/Funding challenges; C7, Social challenges.

Step 5: The influence and relation score of the main groups of challenges and sub-challenges. This step involves row and column summing of the total-relation matrix. The total row $R_i$ and total column $C_i$ are calculated using Equations (7) and (8), respectively. Tables 9 and 10 depict the row and column summation for each main challenge and their sub-challenges, as well as how those challenges are categorized into cause-and-effect groups depending on the strength of their relationships $(R_i - C_i)$. Positive results from $(R_i - C_i)$ indicate a challenge with the cause, while negative results indicate an effect.

$$R_i = \sum_{j=1}^{n} t_{ij} \tag{7}$$

$$C_i = \sum_{i=1}^{n} t_{ij} \tag{8}$$

**Table 9.** The influence and relation score of the main groups of challenges.

| Criteria (Main Groups of Challenges) | $R_i$ | $C_i$ | $R_i + C_i$ | $R_i - C_i$ | Identify |
|---|---|---|---|---|---|
| Supply Chain Management Challenges | 1.238 | 1.101 | 2.339 | 0.137 | Cause |
| Regulation and Policies Challenges | 1.490 | 1.349 | 2.839 | 0.140 | Cause |
| Industry Challenges | 1.518 | 1.418 | 2.936 | 0.100 | Cause |
| Organization's Internal challenges | 1.420 | 1.449 | 2.869 | −0.029 | Effect |
| Technology and Operational Challenges | 1.413 | 1.448 | 2.861 | −0.035 | Effect |
| Economic/Funding Challenges | 1.312 | 1.279 | 2.591 | 0.033 | Cause |
| Social Challenges | 1.068 | 1.415 | 2.483 | −0.347 | Effect |

**Table 10.** The influence and relation score of the sub-challenges for each main group of challenges.

| Main Groups of Challenges | Sub-Challenges | $R_i$ | $C_i$ | $R_i + C_i$ | $R_i - C_i$ | Identify |
|---|---|---|---|---|---|---|
| Supply Chain Management Challenges | S1 | 13.09 | 12.70 | 25.79 | 0.38 | Cause |
|  | S2 | 13.20 | 12.70 | 25.90 | 0.50 | Cause |
|  | S3 | 13.19 | 12.89 | 26.08 | 0.30 | Cause |
|  | S4 | 12.21 | 12.24 | 24.45 | −0.03 | Effect |
|  | S5 | 5.89 | 7.05 | 12.94 | −1.15 | Effect |
| Regulation and Policies Challenges | R1 | 9.99 | 8.36 | 18.35 | 1.63 | Cause |
|  | R2 | 9.16 | 8.36 | 17.52 | 0.80 | Cause |
|  | R3 | 7.96 | 8.77 | 16.73 | −0.81 | Effect |
|  | R4 | 8.36 | 9.97 | 18.34 | −1.61 | Effect |
| Industry Challenges | I1 | 11.13 | 12.59 | 23.72 | −1.46 | Effect |
|  | I2 | 10.80 | 11.99 | 22.79 | −1.20 | Effect |
|  | I3 | 11.97 | 11.70 | 23.67 | 0.28 | Cause |
|  | I4 | 12.30 | 9.92 | 22.22 | 2.38 | Cause |

**Table 10.** *Cont.*

| Main Groups of Challenges | Sub-Challenges | $R_i$ | $C_i$ | $R_i + C_i$ | $R_i - C_i$ | Identify |
|---|---|---|---|---|---|---|
| Organization's Internal challenges | O1 | 11.21 | 10.71 | 21.92 | 0.51 | Cause |
| | O2 | 10.44 | 10.96 | 21.40 | −0.52 | Effect |
| | O3 | 11.21 | 10.71 | 21.92 | 0.51 | Cause |
| | O4 | 10.96 | 11.46 | 22.42 | −0.50 | Effect |
| Technology and Operational Challenges | T1 | 11.32 | 11.69 | 23.01 | −0.37 | Effect |
| | T2 | 10.97 | 12.13 | 23.10 | −1.16 | Effect |
| | T3 | 9.98 | 10.04 | 20.03 | −0.06 | Effect |
| | T4 | 11.83 | 10.80 | 22.63 | 1.03 | Cause |
| | T5 | 10.73 | 10.76 | 21.49 | −0.04 | Effect |
| | T6 | 10.81 | 10.21 | 21.02 | 0.59 | Cause |
| Economic/Funding Challenges | E1 | 4.53 | 5.02 | 9.56 | −0.49 | Effect |
| | E2 | 4.68 | 5.02 | 9.70 | −0.35 | Effect |
| | E3 | 5.70 | 5.97 | 11.67 | −0.27 | Effect |
| | E4 | 5.74 | 5.54 | 11.28 | 0.20 | Effect |
| | E5 | 5.87 | 4.97 | 10.84 | 0.90 | Cause |
| Social Challenges | SO1 | 9.85 | 9.85 | 19.70 | 0.00 | N/A |
| | SO2 | 10.48 | 9.48 | 19.96 | 0.99 | Cause |
| | SO3 | 10.18 | 9.48 | 19.66 | 0.70 | Cause |
| | SO4 | 9.50 | 11.14 | 20.64 | −1.64 | Effect |
| | SO5 | 10.17 | 10.22 | 20.39 | −0.05 | Effect |

Step 6: Producing a causal diagram (Impact Relationship Map).

Based on the total relationship matrix, a cause-and-effect digraph was developed to illustrate the links between the different main challenge groups. Furthermore, another cause-and-effect digraph constructed for each main group sub-challenges. Figures 3 and 4 show the causal relationship between the main group of challenges and their sub-challenges within the main group of challenges, respectively.

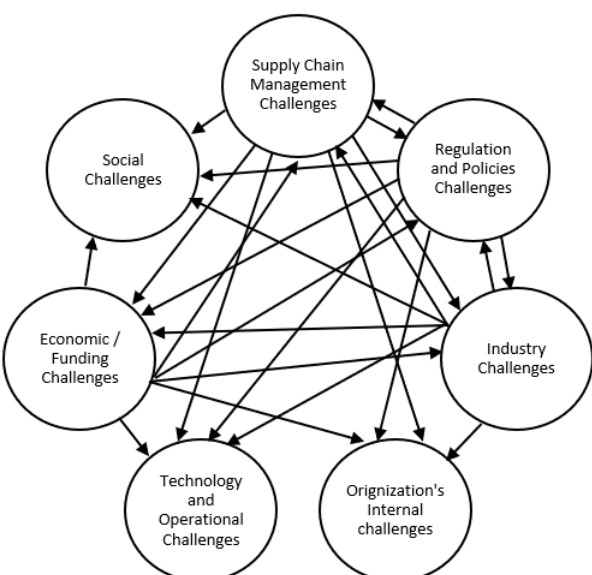

**Figure 3.** Impact relationship map for the main groups of challenges.

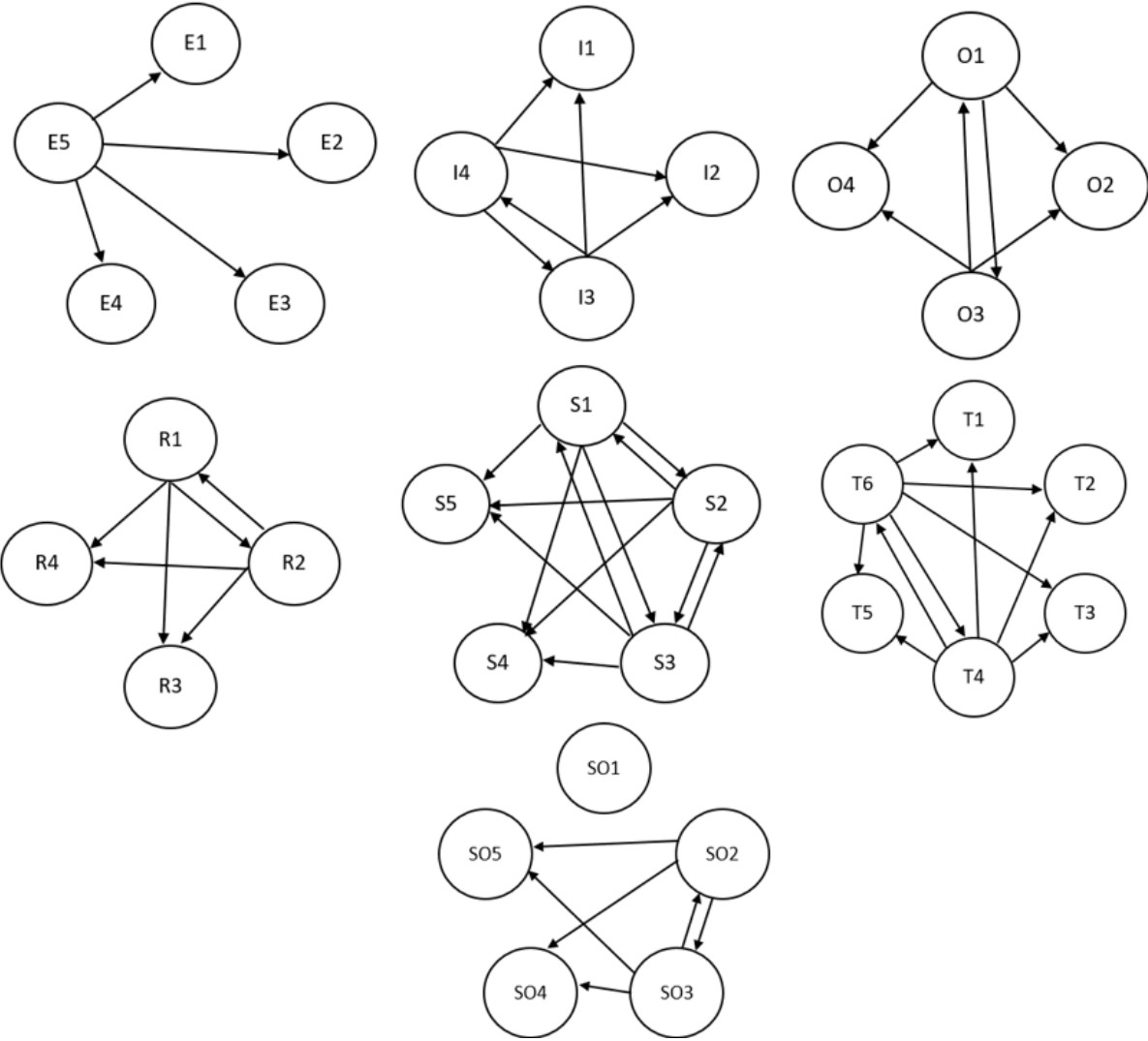

**Figure 4.** Impact relationship map for the sub-challenges for each main group.

Step 7: Ranking the main groups of challenges and their sub-challenges.

Equations (9) and (10) were used to calculate the weights for each main group of challenges and their sub-challenges, as recommended by Vinodh and Wankhede [51]. The local weight of each sub-challenge was multiplied by the weight of the corresponding main group to determine the global weight of each challenge. The overall weight and ranking are displayed in Table 11. Equation (10) gives the weights of each main criterion and their sub-criterion where $\sum_{i=1}^{n} W_i = 1$

$$L_i^{average} = \frac{(R_i + C_i) + (R_i - C_i)}{2} = \sum_{j=1}^{n} t_{i,j} \tag{9}$$

$$W_i = \frac{L_i^{average}}{\sum_{i=1}^{n} L_i^{average}} \tag{10}$$

**Table 11.** Challenges ranking.

| # | Main Challenges Group | Group Weight | # | Sub-Challenge Code | Local Weight | Local Ranking | Global Weight | Global Rank |
|---|---|---|---|---|---|---|---|---|
| 1 | Supply Chain Management Challenges | 0.131 | 1 | S1 | 0.2272 | 3 | 0.0298 | 18 |
| | | | 2 | S2 | 0.2292 | 1 | 0.0300 | 14 |
| | | | 3 | S3 | 0.2291 | 2 | 0.0300 | 16 |
| | | | 4 | S4 | 0.2121 | 4 | 0.0278 | 19 |
| | | | 5 | S5 | 0.1024 | 5 | 0.0134 | 33 |
| 2 | Regulation and Policies Challenges | 0.157 | 6 | R1 | 0.2816 | 1 | 0.0443 | 1 |
| | | | 7 | R2 | 0.2582 | 2 | 0.0407 | 4 |
| | | | 8 | R3 | 0.2244 | 4 | 0.0353 | 12 |
| | | | 9 | R4 | 0.2358 | 3 | 0.0371 | 10 |
| 3 | Industry Challenges | 0.160 | 10 | I1 | 0.2409 | 3 | 0.0387 | 5 |
| | | | 11 | I2 | 0.2337 | 4 | 0.0375 | 9 |
| | | | 12 | I3 | 0.2592 | 2 | 0.0416 | 3 |
| | | | 13 | I4 | 0.2662 | 1 | 0.0427 | 2 |
| 4 | Organization's Internal challenges | 0.150 | 14 | O1 | 0.2559 | 1 | 0.0384 | 6 |
| | | | 15 | O2 | 0.2382 | 4 | 0.0358 | 11 |
| | | | 16 | O3 | 0.2559 | 1 | 0.0384 | 6 |
| | | | 17 | O4 | 0.2501 | 3 | 0.0375 | 8 |
| 5 | Technology and Operational Challenges | 0.149 | 18 | T1 | 0.1725 | 2 | 0.0258 | 21 |
| | | | 19 | T2 | 0.1671 | 3 | 0.0250 | 22 |
| | | | 20 | T3 | 0.1521 | 6 | 0.0227 | 30 |
| | | | 21 | T4 | 0.1802 | 1 | 0.0269 | 20 |
| | | | 22 | T5 | 0.1634 | 5 | 0.0244 | 25 |
| | | | 23 | T6 | 0.1646 | 4 | 0.0246 | 23 |
| 6 | Economic/Funding Challenges | 0.139 | 24 | E1 | 0.1710 | 5 | 0.0237 | 26 |
| | | | 25 | E2 | 0.1763 | 4 | 0.0245 | 24 |
| | | | 26 | E3 | 0.2149 | 3 | 0.0298 | 17 |
| | | | 27 | E4 | 0.2164 | 2 | 0.0300 | 15 |
| | | | 28 | E5 | 0.2213 | 1 | 0.0307 | 13 |
| 7 | Social Challenges | 0.113 | 29 | SO1 | 0.1963 | 4 | 0.0222 | 31 |
| | | | 30 | SO2 | 0.2088 | 1 | 0.0236 | 27 |
| | | | 31 | SO3 | 0.2029 | 2 | 0.0229 | 28 |
| | | | 32 | SO4 | 0.1893 | 5 | 0.0214 | 32 |
| | | | 33 | SO5 | 0.2027 | 3 | 0.0229 | 29 |

## 4. Results and Discussion

### 4.1. Supply Chain Management Challenges

This category comes in sixth place in terms of overall weight and contributes (13.1%) to the total weight. The sub-challenges under this group are uncertainty in return flows (S1), lack of consideration for a CE supply chain (S2), inefficient resource utilization (S3), lack of flexibility in implementation of CE phases (S4), challenges in taking back products (4Rs) (S5). From Table 11, the order of the priority is revealed to be: S2 > S3 > S1 > S4 > S5. The challenges S1, S2 and S3 represent the cause, while others S4 and S5 are the effect. The most impactful challenges are S2, S3, S1 with local ranking 1, 2, 3, respectively. The global ranking of S2, S3, S1 are 14, 16, 18, respectively, out of 33 total challenges. The effect challenges S4 and S5 have local rankings of 4, 5 respectively. The global ranking for S4 is 19 out of 33 challenges, and S5 comes at the last ranking place of 33. The lack of consideration for a CE supply chain (S2) appears to be a common issue in the supply chain. If a CE supply chain is not considered, its effects compound and present numerous symptoms. Challenges in taking back products (S5) and lack of flexibility in implementation of CE phases (S4) are the respective symptoms. It is difficult for hospitals to take back used products if the supply chain at its core does not support the CE model. Furthermore,

inefficient resource utilization (S3) makes it logistically difficult for HCW practices to shift to a more sustainable approach.

### 4.2. Regulation and Policies Challenges

This category comes in as the second highest weight and contributes (15.7%) of the total weight. The sub-challenges of this group are lack of governmental legislation on CE healthcare policies (R1), lack of sustainability foundation practices, incentives, and policies (R2), lack of R&D to reinforce sustainability practices and create/update (R3), and lack of standards for existing CE (R4). From the calculation in Table 11, the order of the priority is shown to be: R1 > R2 > R4 > R3. The challenges R1 and R2 represent the cause, while others R4 and R3 are the effect. The most influential challenge is R1 with a local ranking of 1. However, the least influential one is R3 with local ranking of 4. Furthermore, R1 has the highest influence among other challenges with a global ranking of 1. This group of challenges carries a significant amount of the overall barrier weight. At the core of HCW practices in the United States is the regulation and policies that shape the practices themselves. The lack of incentive and legislation in turn results in minimal investment into more sustainable practices. Furthermore, the creation of standards for a CE that can be assessed nationally would aid in standardizing what it takes to be sustainable.

### 4.3. Industry Challenges

This category has the highest weight and contributes (16%) among all other main groups. This group contains 4 sub-challenges which are: resistance to improve existing practices to be sustainable (I1), minimal knowledge on training and expertise for sustainability (I2), lack of realistic CE business model (I3), and lack of consumer interest in the environment (I4). The sub-challenges order of the priority appears to be: I4 > I3 > I1 > I4. Here, I4 has the most influence following it with I3. It is worth nothing their global ranking is within the top ranking of 2 and 3, respectively. Systemic transformation is deep rooted in the industry's philosophy. I3 and I4 represent robust barriers to the shift to a CE. Consumers represent the core income of the industry. If the consumers lack an interest in sustainable practices, then there is a clear resistance to improving existing practices. Furthermore, the absence of a realistic CE business model makes it difficult for consumers to even consider transforming their practices. Further research should be done to develop and understand a CE business model that is applicable nationally. In addition, it should provide financial incentives to prove its capabilities.

### 4.4. Organization's Internal Challenges

This category comes in third place and contributes (15%) of the total weight. The sub-challenges that fall under this group are carelessness in usage (O1), poor contribution by stakeholders (O2), strict requirements for advanced education and training (O3), and conflict of interests among departments (O4). The order of the sub-challenges priority is demonstrated to be: O1 = O3 > O4 > O2. The two sun-challenges O1 and O2 had the same priority level. The ranking of O1 and O3 are the same with local weight (0.255), that makes both have the local ranking of 1 repeatedly. Moreover, they share the same global ranking of 6. In the modern era, consumerism is at an all-time high, and thus, there is a lack of accountability in excessive use of medical products. Organizations should consider creating standards and incentives to prevent the excessive or careless use of products. In addition, the strict requirements for advanced education and training make it difficult for someone to work in the HCW management field. These requirements result in a lack of employees available who are knowledgeable. O4 and O2 represent the organization's political turmoil in shifting to more sustainable practices.

### 4.5. Technology and Operational Challenges

Another category with (14.9%) weight contribution. This main group of challenges has the highest number of sub-challenges. The six sub-challenges are lack of environmen-

tally friendly disposal practices (T1), challenges in reprocessing (T2), insufficient product traceability (T3), the associated complexity with circular economy practices (T4), lack of key performance indexes to adequately measure "green" performance of hospitals (T5), and minimal advocation to use medical informatics to reinforce sustainability (T6). From Table 11, the order of the priority is displayed to be: T4 > T1 > T2 > T6 > T5 > T3. T4 and T6 are the cause challenges while the others are effect. T4 has the first local ranking since it is the most influential sub-challenge in this group. However, the least significant one is T3 with a global ranking of 30. Technology shapes the way certain practices are carried out. Since a CE in HC, particularly in the United States, has not been researched thoroughly, there is a complexity associated with its practices. Minimal advocation to use medical informatics also creates operational challenges in assessing how sustainable an organization is. Currently, the technology and infrastructure provide minimal incentive to be more "green" given its technological and logistic difficulty.

### 4.6. Economic/Funding Challenges

This category comes in fifth place with (13.9%) weight contribution. The economic/funding challenges have five sub-challenges, and they are low eco-efficiency of technological processes (E1), conflict of interests and poor communication among departments (E2), high costs associated with circular economy disposal practices (E3), unwillingness to invest more (E4), and inadequate allocation of funds (E5). The order of the priority is shown to be: E5 > E4 > E3 > E2 > E1. E5 represents the only cause sub-challenge within this group. The local ranking and global ranking of E5 are 1, 13, respectively. On the other hand, E1 with local weight (0.171) has the lowest local ranking of 5. With the inadequate allocation of funds (E5) comes an abundance of economic challenges. There is simply an unwillingness to invest more in CE practices due to its high initial costs. As a result, funds are not allocated in such a way that incentivize and support CE practices.

### 4.7. Social Challenges

This category comes in the last place with the lowest (11.3%) weight contribution among all other main groups. The last main group has five sub-challenges, and they are cultural barriers (SO1), minimal public incentives and interest regarding sustainability practices (SO2), lack of awareness about CE practices and resources (SO3), the widespread use disposal of medical products/devices/supplies (SO4), and the lack of environmental impact awareness on public health (SO5). From Table 11, the order of the importance is shown to be: SO2 > SO3 > SO5 > SO1 > SO4. Here, SO2 and SO3 are the caused sub-challenges with local weights (0.208) and (0.202), respectively. Hence, make them the two top local ranking, SO1 indicates a zero value in step 5, which means inconclusive. Thus, more opinions will determine whether this criterion should be considered. In the United States, adequate waste management practices are not deeply rooted into society. A potential solution for the coming years would be to educate the younger generation on more sustainable waste management practices. Simply creating awareness and providing the resources on how to be more sustainable can provide benefits. It is essential that the waste management culture is improved to ensure a more sustainable economy.

## 5. Conclusions

Although attractive, the implementation of a CE in HCW requires systemic transformation and cooperation between all stakeholders. A CE that is built on foundations of resource efficiency, energy conservation, and the 4Rs provides an alternative to the unsustainable current practices. This study identified the challenges to the implementation of a CE in HCW management. The methodology validated the challenges by the aid of expert opinion and ranked the weights of the challenges using the DEMATEL methodology. The approach also outlined significant cause and effect relationships between main challenges and their respective sub-challenges. The research is limited by a small sample size of HCW management experts limited to the context of the Miami, United States area. In the future,

a larger sample size can be assessed nationwide to gain deeper insight, and an assessment of both developing and developed HCW management systems. Despite the limitation, the research will provide context for future studies into the implementation of a CE. In addition, the research will help organizations in addressing the problems that may arise in HCW management. From a managerial perspective, simply understanding the root causal barriers to sustainability can provide a tremendous benefit. In addition, it can aid in the creation of more sustainable practices. The research found that industry transformation requires novel and realistic business models that are incentivized by complimentary regulations.

**Author Contributions:** Y.A. developed the qualitative portion of this study. Y.A. and A.M. developed the main challenges and sub-challenges tables. A.M. developed the methodology, results, and discussion sections. V.O. supervised the overall research. All authors have read and agreed to the published version of the manuscript.

**Funding:** This research received no external funding.

**Institutional Review Board Statement:** Not applicable.

**Informed Consent Statement:** Informed consent was obtained from all subjects involved in the study.

**Data Availability Statement:** Not applicable.

**Acknowledgments:** The authors would like to thank some of the local healthcare disposal practice experts for their insight and guidance.

**Conflicts of Interest:** The authors declare no conflict of interest.

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
