# Peer review of "Healthcare Waste and Sustainability: Implications for a Circular Economy"

_sustainability, doi:10.3390/su15107788_

Round 1

Reviewer 1 Report

This contribution (sustainability-2253497) studies healthcare waste and sustainability, emphasizing the implications for a circular economy. While attractive in some ways, this manuscript needs to present a careful review of the literature. 

The document has no logical progression; the manuscript sections seem disconnected. While interesting, the manuscript is written more like a description than a scientific examination of the subject. The latter is the major shortcoming.

There needs to be more quality and presentation of the information. Figures and Tables must be improved.

The authors' affiliations need to be clarified.

Author Response

Hello.

Thank you so much for taking the time to read our paper and provide your feedback. We really appreciate it.

We added some further context and analysis into our results and discssion section to let the material flow more naturally. And, to examine the results themselves in the context of the study.  

In addition, we sorted out the affiliations in accordance to the MDPI format. 

Please see attached the revised version.

Thank you so much again,

Kind regards.

Reviewer 2 Report

My comments are as follows:

1) Please be careful to properly revise the manuscript according to MDPI format: headings of tables and figures, the position of figures, Table colors, etc. Moreover, references should be formatted according to MDPI style.

2) The affiliations of the second and third authors are missing. Also, the term "Affiliation 1;;" should be deleted. Moreover, academic emails should be provided for all authors.

3) There are many repetitive contents on pages 8 and 9 with the same heading numbers and the same content. I wonder about the accuracy of the contents.

4) There are some missing references such as "Dynamic demand-centered process-oriented data model for inventory management of hemovigilance systems, Healthcare Informatics Research 2021" or "Multivariate time-series blood donation/demand forecasting for resilient supply chain management during COVID-19 pandemic, Cleaner Logistics and Supply Chain 2022", which can be added into the literature review section.

5) Managerial insights should be added to show how the outcomes of this manuscript can help decision-makers.

Author Response

Hello.

Thank you so much for taking the time to read and review our paper. We really appreciate it.

1) We revised the paper in accordance to MDPI format.

2) We have sorted out the affiliations as well.

3) The contents on page 8 and 9 provide context to the barriers themselves.

4) Thank you for providing these papers. We included "Dynamic demand-centered process-oriented data model for inventory management of hemovigilance systems, Healthcare Informatics Research 2021" in our literature review.

5) We added managerial insights to the conclusion.

Please see attached the revised manuscript.

Thank you so much again,

Reviewer 3 Report

The reviewed article addresses very important issues related to the sustainable management of waste generated in healthcare facilities. The introduction is interesting and clearly explains the need to undertake research on the transition from linear economy to circular economy. The authors conducted a thorough review of the literature related to the management of waste from the healthcare system, thanks to which they identified the challenges faced by the introduction of CE. The methodology adopted in the work has been described in a comprehensive manner. Despite these positive comments, the article requires a major revision before being published in Sustainability.

1.      The article needs to be adapted to the requirements of the journal.

2.      Tables 6 to 11 (number 2 in the text - line 430) require commentary and should be included in the chapter Results and discussion. The same remark applies to Figures 4 and 5.

3.      The presented conclusions do not actually refer to the obtained results. Please rewrite them.

4.      Chapter 4 is titled Results and discussion. However, there is no discussion in this chapter. The authors present their results there, without explaining (commenting) them in any way.

5.      Chapter 4 - please correct the numbering of subsections. Only 4.1 is valid.

6.      Line 49, 54, 75, 439 - too many spaces

7.      Lines 263-264 - please combine the two parts of the sentence

8.      Line 321 - half of the table title is in bold, the other half is not.

Author Response

Hello,

Thank you so much for taking the time to read and review our paper. We really appreciate it.

We took into consideration all your comments. The article has been modified in accordance to MDPI format. All extra spaces and tables have been fixed. For chapter 4, we added discussion to the results to assess the implications in the context of this research. As for the chapter 4 formatting and numbering, could you please check it again to see if its alright?

Please see attached the revised manuscript. 

Thank you so much again,

Kind regards.

Reviewer 4 Report

It's an interesting work analysis of different perspective to encourage the use of new technologies for healthcare waste and change the current way of the final disposition.

Author Response

Hello,

Thank you so much for taking the time to read and review our paper. We really appreciate it.

Kind regards.

Round 2

Reviewer 1 Report

The manuscript sustainability-2253497 has been improved. However, below 50% of the cited references were published recently (after 2020); it is necessary to add recent references.

Tables must be edited and presented in a uniform format.

The authors' affiliations still need to be clarified.

Author Response

Thank you so much for time and feedback.

Tables are now in a uniform format, and authors' affiliations now is clarified.

For the references part, 21 of references are most recent ones out of 51 references. 

Reviewer 2 Report

The revised version can be accepted.

Author Response

Thank you so much for time , kindly find the new revised version.

Reviewer 3 Report

The authors have corrected the article according to my suggestions and I believe it can be published in Sustainability.

Author Response

Thank you so much for time and feedback,

Round 3

Reviewer 1 Report

The authors have improved their manuscript and responded to the points indicated.